# SEMANTIC, OPTIMAL, LOSSLESS VECTOR QUANTIZATION

## ABSTRACT

Is it possible to derive an *optimally compact* image representation that *preserves semantic information without performance loss* for a class of downstream tasks? This paper addresses this fundamental question by providing a formal definition of semantic lossless optimal compression. We introduce a framework **S**emantic **O**ptimal **LO**ssless Vector **Q**uantization (**SOLO-VQ** as a practical realization to address this concept. Unlike prior works, which often rely on heuristics and evaluate on generic image datasets where optimality is unverifiable, we propose a novel evaluation protocol. We construct a series of synthetic datasets and associated tasks where the information-theoretic rate limits for lossless compression are computable. Within these controlled environments, we empirically demonstrate that SOLO-VQ achieves provably optimal and lossless compression, effectively reaching the theoretical lower bounds. Our work establishes a principled foundation for goal-oriented semantic media data compression and suggests a promising methodology towards achieving this goal for compressive real-world image transmission.

## 1 INTRODUCTION

Learning compact, interpretable, and task-relevant encoding from high-dimensional inputs such as images is a fundamental goal in machine learning. Classical approaches Ballé et al. (2016; 2018); Mentzer et al. (2020); Cheng et al. (2020); He et al. (2022) to compression often optimize for low-level fidelity—preserving pixel values or human-level perception-without explicit regard for the semantic utility of the compressed representation. In contrast, we advocate for a new perspective: *semantic compression*, which seeks to discretize visual data into compact codes sufficiently allow the receiver to solve a predefined set of semantic tasks without performance loss.

We formalize semantic compression as a framework where a compressor produces a discrete representation $C = E(X)$ from an image $X$, such that downstream semantic tasks $\{T_i(X)\}$ can be accurately predicted from $C$. This objective defines a notion of *lossless semantic compression*, wherein the semantic information required for the tasks is fully preserved in the code. We further define the notion of *optimality*, where the entropy of the code matches the joint entropy of the task labels, providing an information-theoretic bound on compression.

To operationalize this framework, we introduce a three-stage training pipeline: (1) learning a latent space for codebook initialization; (2) constructing discrete codebooks via k-means clustering in latent space; and (3) introducing projectors to further adapt initialized codebooks to the latent space for tasks' performance. This disentangled training process enables precise control over semantic accuracy and code length, allowing us to analyze the trade-off between compression and task performance.

While prior vector quantization methods such as VQ-VAE van den Oord et al. (2017) and VQ-GAN Esser et al. (2021) have demonstrated strong performance in generative modeling and reconstruction, they primarily focus on preserving appearance rather than semantic content. Our approach draws inspiration from their discretization mechanisms but repurposes them with a fundamentally different goal: preserving semantic information rather than reconstructing pixels.

Our formulation is also complementary to the broader landscape of representation learning methods, which often rely on proxy objectives (e.g., contrastive loss, masked prediction) and heuristic

inductive biases. Semantic compression, in contrast, is defined directly in terms of its end use: solving tasks. This makes the learned representations inherently interpretable and their utility directly measurable.

To rigorously evaluate semantic compression, we construct synthetic datasets with known task structures and computable entropies. These controlled environments allow us to empirically verify losslessness and optimality, providing a stepping stone toward applying semantic compression in real-world settings.

## 2 SEMANTIC OPTIMAL LOSSLESS COMPRESSION

In this section, we formally define semantic compression, its lossless form, and its optimal lossless form.

### 2.1 DEFINITION: SEMANTIC COMPRESSION

We formally define semantic compression of images as comprising the following components:

1. A pretrained semantic compressor $E$ that takes as input an image, represented by a random variable $X$, and produces a discrete and compact semantic code $C = E(X)$.

2. A set of downstream semantic tasks $\mathcal{T} = \{T_1, \ldots, T_m\}$, where each $T_i$ is a function of the input variable $X$, such as image segmentation, depth estimation, semantic understanding, image classification, or object counting.
   A set of downstream semantic tasks $\mathcal{T} = \{T_1, \ldots, T_n\}$, where each $T_i$ is a function of the input variable $X$, and $T_i(X)$ provides the ground-truth target for $X$ under the $i$-th task—for example, a segmentation map, a depth map, or a classification label.

3. A set of task-specific decoding heads $\mathcal{H} = \{H_1, \ldots, H_m\}$, each fine-tuned on its respective task in $\mathcal{T}$. Each decoder $H_i$ takes the semantic code $C$ as input and produces a prediction $H_i(C)$ for task $T_i$.

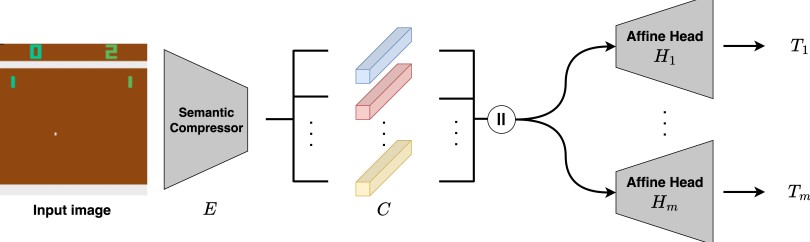

Figure 1: An Experimental Proof-of-Concept for Semantic Compression.

At first glance, this formulation resembles the general verification of unsupervised representation learning(Le et al., 2012; Chen et al., 2020; Grill et al., 2020; Caron et al., 2021). The key distinction is that semantic compression does not rely on inductive biases from heuristic architectures, such as assuming semantic invariance across augmented views(Chen et al., 2020; He et al., 2020) or enforcing distant representations from different samples(Chen et al., 2020; He et al., 2020; Wu et al., 2018) or patches(Caron et al., 2021; Wang et al., 2021; Chen et al., 2023). Instead, semantic compression is directly optimized with respect to a predefined set of downstream tasks with an information bottleneck. This makes the framework more interpretable and allows the compactness of the representation to be explicitly considered by design.

Although semantic compression may appear less general than representation learning—since it learns an encoder optimized for a specific set of tasks—this is not necessarily a limitation. In practice, semantic signals in images are inherently defined by a wide range of semantic tasks, many of which share significant information (e.g., semantic segmentation and depth estimation). Therefore, we expect that a semantic encoder trained on a sufficiently broad set of tasks can capture rich semantic content and generalize to unseen downstream tasks.

## 2.2 DEFINITION: SEMANTIC LOSSLESS COMPRESSION

A semantic compressor $E$ is said to be *lossless* if, for any input sample $X$, the resulting code $C = E(X)$ allows the construction (i.e., training) of a set of decoding heads $\mathcal{H} = \{H_1, \ldots, H_m\}$ such that all task predictions are exactly correct, that is,

$$H_i(C) = T_i(X), \quad \forall i \in \{1, \ldots, n\}, \forall X.$$

In practice, we adopt a relaxed version of this condition: if, for a well-partitioned test set with no data leakage, the codes produced by $E$ enable us to train decoding heads $\mathcal{H}$ that achieve perfect accuracy on all tasks within a reasonable amount of time, we consider the compressor $E$ to be lossless.

In practice, we relax this condition: if the codes produced by $E$ on a well-partitioned, leakage-free test set allow us to train a set of decoding heads $\mathcal{H}$ that achieve perfect accuracy within a reasonable amount of time, we consider $E$ to be lossless. In the general lossy case, we interpret the loss on semantic tasks as the *distortion* induced by the compression.

## 2.3 DEFINITION:SEMANTIC OPTIMAL LOSSLESS COMPRESSION

A lossless semantic compressor $E$ is said to be *optimal* if the entropy of its output code $C$ equals the joint entropy of the downstream tasks, i.e.,

$$H(C) = H(T_1(X), \ldots, T_m(X)).$$

This joint entropy represents the information-theoretic lower bound for lossless compression, determining the minimal achievable average code length under a lossless semantic representation.

For a non-optimal compressor, one can consider its *code length redundancy* relative to the optimal code length $H(C) - H(T_1(X), \ldots, T_m(X))$.

Alternatively, one may directly analyze the trade-off between the compression rate $H(C)$ and the average distortion incurred on downstream tasks.

## 3 PUSHING TO THE LIMIT

An ultimate goal is to obtain a semantic compressor that approaches the performance of an optimal lossless compressor, by achieving near-minimal distortion and near-optimal code length on real-world image datasets and a universal set of downstream tasks. However, the entropy of real-world data and tasks is often intractable to estimate accurately due to their inherent complexity and the limited number of available samples. Moreover, verifying perfect prediction for a wide range of downstream tasks would require an enormous amount of human annotation and validation. As a result, it is infeasible to directly measure the loss and code length redundancy of a semantic encoder on realistic datasets and task collections.

As a first step toward this ambitious objective, we aim to validate our approach in a toy environment where both optimality and losslessness can be explicitly verified. To this end, we introduce a suite of synthetic datasets paired with carefully designed semantic tasks, each possessing the following key properties:

- Each task admits a lossless decoder, which can be trained to achieve perfect prediction;
- The combined entropy $H(T_1(X), \ldots, T_m(X))$ of the tasks is analytically computable.

These two properties enable an exciting possibility: on such synthetic datasets and tasks, it becomes feasible to empirically achieve and verify an *optimal lossless compressor*. In Section 4.1, we demonstrate that our proposed method indeed achieves this goal.

## 3.1 SYNTHESIZED DATASETS, TASKS AND EVALUATION METRICS

We design two categories of synthetic datasets: a **Pong-style image dataset** and a **geometric object arrangement (Geo) datasets**.

- **Pong Dataset:** We use Python scripts to generate images resembling those in the Atari Pong game. Each image includes a ball, a score pair, and two paddles for both players, and score indicators. Importantly, this dataset is not derived from the actual game engine and does not reflect the true game dynamics or its associated distribution. The ball and paddles are restricted to a finite set of discrete grid positions; scores are integers between 0 and 3; and the two background regions are independently colored from a predefined discrete color set. This dataset is designed to simulate the type of visual environments commonly encountered by AI agents in game-like settings. Appendix provides illustrations of the synthesized datasets, including Pong-S 6, Pong-SPB 7, and Pong-SPC 8.

- **Geo Datasets:** We use Blender, combined with scripted automation, to generate images in which a set of geometric objects are randomly arranged on a gray background. Each object is selected from a finite set of shapes and colors, and its position is restricted to a discrete grid. This dataset is designed to simulate the visual recognition setting in 3D scene understanding. Appendix provides illustrations of the synthesized datasets, including Geo-4Seg 3 and Geo-6Seg 4.

The discrete nature of states in both datasets allows for tractable analysis and exact enumeration. For simplicity, we primarily sample all configurations uniformly across their possible state spaces. Nonetheless, we also explore the impact of non-uniform data distributions on encoding performance.

Across these two datasets, we consider the following types of semantic tasks and corresponding evaluation metrics:

- **Information Extraction (Classification) Tasks:** For example, predicting paddle positions or scores from Pong images. Prediction accuracy is the metric used to evaluate this type of tasks.

- **Semantic Segmentation Tasks:** Segmenting geometric objects from the background in the object arrangement dataset, with optional grouping of same-shaped objects into identical segmentation masks. Two standard metrics are used to evaluate segmentation tasks:

  - **Mean Intersection over Union(mIoU)** is defined as the average Intersection over Union(IoU) across all classes, where IoU for each class is computed as the ratio between the intersection and the union of the predicted and ground-truth regions.
  - **Mean Class Accuracy** is defined as the average of per-class accuracies, where the accuracy for each class is computed as the ratio of correctly predicted pixels to the total number of pixels in that class.

More detailed descriptions of the dataset and tasks are provided in the experiments section 4, along with example visualizations included in the appendix.

## 3.2 Proposed Method

We propose a three-stage approach **SOLO-VQ** to obtain a semantic lossless optimal compressor, as shown in Fig.2. The **semantic compressor** $E$ consists of two main components: **An encoder** $\mathcal{E}$ that generates continuous representation vectors $\{v_i\}_{i=1}^{n}$ for an input image $X$ and **a set of vector quantizers** $\{Q_i\}_{i=1}^{n}$ that correspondingly quantize $\{v_i\}_{i=1}^{n}$ to $n$ tokens as the discrete semantic code. The $m$ **affine heads** $\{H_i\}_{i=1}^{n}$ are then used to decode the code into the targets of the $m$ downstream tasks $\{T_i\}_{i=1}^{m}$.

**Stage 1: VQ-Aware Latent Pretraining.** In this stage, we pretrain the encoder, the affine heads, and the dimension reduction/expansion modules in an end-to-end manner without quantization. The goal is to prepare a well-structured latent space suitable for subsequent vector quantization. A key intuition is that affine heads partition the input space into regions bounded by hyperplanes, where each region maps to the same output. Vector quantization similarly partitions the latent space into *Voronoi cells*, which are also defined by hyperplane boundaries. This structural similarity implies that affine heads encourage the latent space to align better with the quantization boundaries, improving overall quantization efficiency.

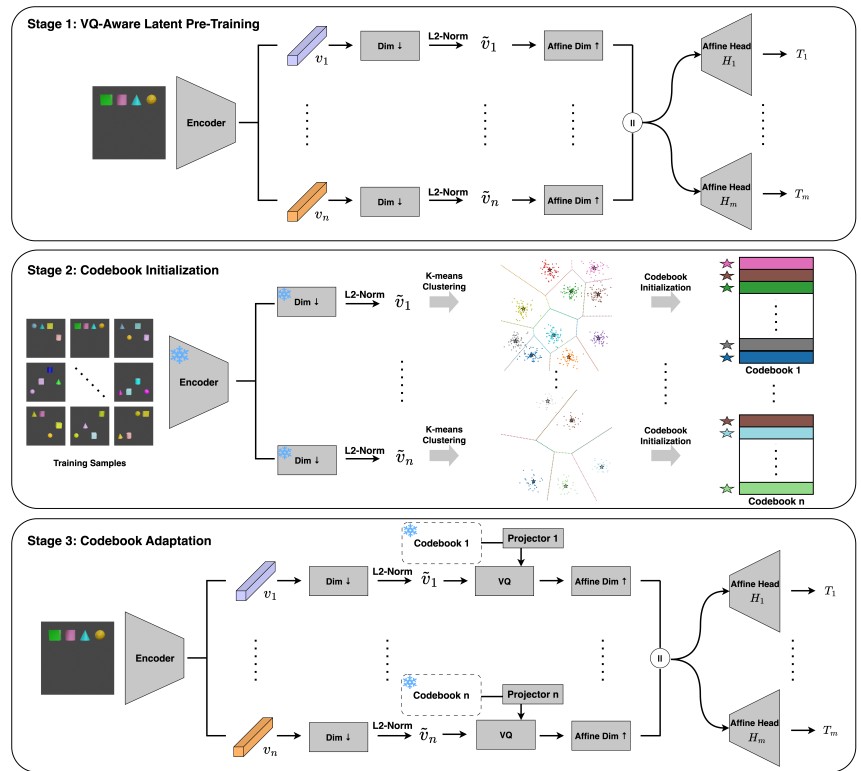

Figure 2: The proposed Semantic Compression method.

**Stage 2: Codebook Initialization.** We freeze the encoder and the dimension reduction components, and perform $\{k_i\}_{i=1}^{n}$-means clustering to initialize either a single shared codebook or $n$ independent codebooks for each latent vector position. Prior work has shown that clustering-based initialization leads to better codebook utilization. The use of multiple independent codebooks allows us to fully leverage the position-specific structure of the latent representation, and later enables progressive codebook shrinking to exploit compression limits. All vectors are quantized in a normalized, reduced-dimensional space, which has also been shown to improve codebook utilization.

**Stage 3: Codebook Adaptation.** In this stage, the codebooks are frozen, and we train a lightweight projector to remap the fixed codebook into the encoder's latent space, while jointly fine-tuning the encoder, the affine heads, and other components. The objective is to adapt the clustering-derived codebook to the downstream semantic tasks. Using a learnable projector to remap the codebook—rather than updating it directly via straight-through gradients—has also been shown to significantly improve codebook utilization. In order to explore the optimal bitrate on multi-token setting, we initialize and gradually reduce the codebooks' size based on the algorithm 1 2.

**Component Architecture.** Encoder and dimension reduction components: a Vision Transformer (ViT); Projector: a lightweight Transformer; Affine dimension expansion components and Affine heads: an affine layer.(linear layer with bias)

## 4 EXPERIMENTS

We evaluate our method in controlled synthetic environments to verify empirical optimality, rate-distortion behavior, generalization ability, and multi-task performance. Our key findings are organized into four parts.

## 4.1 SEMANTIC OPTIMAL LOSSLESS COMPRESSION

We begin by demonstrating that our method achieves *optimal* and *lossless* semantic compression on two synthetic datasets: Pong image classification and small-scale semantic segmentation. These controlled environments are designed so that all possible semantic configurations can be exactly enumerated, allowing us to compute closed-form entropy bounds. In both settings, our method matches the theoretical lower bound on bitrate while attaining perfect task performance—providing strong empirical evidence of semantic optimality.

**Score Classification on Pong dataset.**   We begin with simple `Pong-S`, where the goal is to extract the score from the pixel-level Pong image. Our method achieves 100% prediction accuracy while using a code length equal to the theoretical minimum entropy $\log_2 16 = 4$ bits.

Table 1: Prediction accuracy on `Pong-S`.

| Stage | Score Pair (%) |
|---|---|
| Pre-Quantization Training | 100.00 |
| Codebook Initialization | 100.00 |

**Small-Scale Semantic Segmentation on Geo dataset.**   We next evaluate the semantic segmentation task on the `Geo-4Seg` dataset. Each image in this dataset contains four distinct geometric shapes arranged in a uniformly random order, resulting in 4! possible shape permutations. Each shape can independently occupy one of three discrete positions, leading to a total of $3^4$ spatial configurations. In total, the dataset contains $4! \times 3^4 = 1944$ unique semantic segmentation maps. To introduce visual variability, the color of each shape is sampled independently and uniformly from a palette of 19 distinct colors. This results in multiple images sharing the same segmentation map but differing in color appearance. The training set includes all 1944 possible segmentation maps, while the test set consists of images with unseen color combinations—that is, none of the color patterns in the test set appear in the training set. We start with a codebook size of 1944, meaning that an optimally lossless compressor must classify inputs exactly and uniformly into all codebook entries.

SOLO-VQ achieves $100\%$ mIoU on all test samples using a codebook size equal to the lower bound 1944. As shown in and Figure.3, significantly outperforming VQ methods such as VQ-VAE and VQ-STE++. The online clustering codebook (CVQ-VAE) can also eventually achieve similar levels performance ($99.93\%$) at the lowerbound. However, CVQ-VAE requires significantly longer time to converge due to its randomness. (see Figure 11).

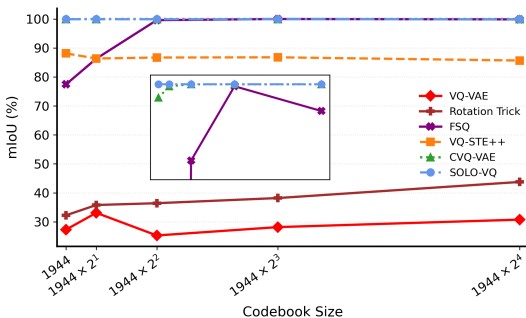

Figure 3: Exploratory comparison of vector-quantization methods in the one-token setting on Geo-4Seg. The x-axis begins at $1,944$, the minimal codebook size dictated by the $1,944$ unique segmentation maps, and doubles at each step ($\times 2$), corresponding to a $+1$-bit increase under uniform usage.

## 4.2 GENERALIZATION IN COMPLEX ENVIRONMENTS

We now test whether semantic compression can scale to more complex distributions and generalize to unseen semantic instances. We use the `Geo-6Seg` dataset, which contains six distinct object types. Each object can independently occupy one of six positions and take on one of nineteen

colors, resulting in a significantly larger number of unique segmentation configurations. ($6! \times 6^6 = 33,592,320$), such that exhaustive enumeration becomes infeasible.

The task entropy lower bound is $\log_2(33,592,320) \simeq 25.002$ bits. Our method achieves $100\%$ test accuracy using only $29.679$ bits of code length, which is remarkably close to the theoretical minimum. Importantly, the test set includes many configurations that were never seen during training, indicating that our model generalizes its semantic abstraction capabilities beyond memorization.

We also plot the rate-distortion curves in Fig. 4, comparing our method with the strongest baselines from Section 4.1. Our method consistently achieves better semantic fidelity at lower code rates, highlighting its compression efficiency and generalization capability.

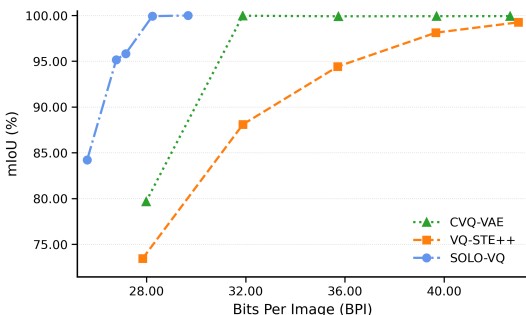

Figure 4: Rate–distortion analysis of vector-quantization methods on Geo-6Seg, plotting rate as bits per image (BPI) versus semantic-segmentation performance (distortion proxy). Each image is represented by four tokens, and the information-theoretic lower-bound rate is 25.002 BPI.

### 4.3 MULTI-TASK COMPRESSION AND COMPOSITIONAL SEMANTICS

To test whether our method supports semantic composition across multiple tasks, we construct a multi-task version of the Pong dataset `Pong-SPC` involving three attributes: **Score Pair (S):** 16 possible combinations; **Background Color (C):** 8 possible combinations; for each score pair, 4 options appear with equal probability; **Paddle Locations (P):** 16 possible combinations (independent of S and C). Visualizations of selected samples can be found in Fig.8. All configurations are sampled with equal probability. The joint entropy of the semantic space can be calculated as:

$$H(S, P, C) = H(P) + H(S) + H(C \mid S) = \log_2 16 + \log_2 16 + \log_2 4 = 12 \text{ bits.}$$

Our method achieves perfect accuracy across all three tasks using exactly 12 bits of code length, demonstrating that it effectively captures both independent and conditionally dependent semantics within one codebook, as shown in Tab.2.

We also evaluated another set of non-independent downstream tasks `Pong-SPB`:Score pair classification (S), Paddle location (P), and Ball location (B). Visualizations of selected samples can be found in Fig.7. Similarly it achieves lossless optimal compression, as shown in Tab.3.

Table 2: Multi-Task score-pair, paddle-location, and background-color classification on `Pong-SPC`.

| Stage | Score Pair (%) | Paddle Location (%) | Background Color (%) |
|---|---|---|---|
| Pre-Quantization Training | 100.00 | 100.00 | 100.00 |
| Codebook Initialization | 100.00 | 100.00 | 100.00 |

Table 3: Multi-Task score-pair, paddle-location, and ball-location classification on `Pong-SPB`.

| Stage | Score Pair (%) | Paddle Location (%) | Ball Location (%) |
|---|---|---|---|
| Pre-Quantization Training | 100.00 | 100.00 | 100.00 |
| Codebook Initialization | 100.00 | 100.00 | 100.00 |

## 4.4 Ablation Study

**Affine Decoder Structure.** We ablate the use of affine decoders in our architecture by replacing them with either MLP decoders with non-linear operations. As shown in Tab.4, we find that the affine decoder (i.e., linear layer with bias) strikes the best code efficiency.

**Codebook Initialization via Clustering.** We ablate the $k$-means-based codebook initialization step in Stage 2 by directly initializing the codebooks randomly and training them end-to-end. As shown in Tab.5, skipping clustering leads to slower convergence, larger quantization errors, and degraded task accuracy. This confirms that initializing the codebook with cluster centroids from latent space provides a strong prior, making quantization more stable and semantically meaningful from the beginning. Further clustering visualizations reveal the impact of affine heads and MLP heads on centroid assignment. MLP heads tend to cause either over-allocation or under-utilization of centroids, as shown in Fig.5.

Table 4: Ablation Study: Comparing non-linear vs. affine designs for the post-latent function in the one-token setting on Geo-4Seg.

| Decoder Function | Pre-Quantization Training | | Codebook Initialization | |
| --- | --- | --- | --- | --- |
| | mean Class Accuracy (%) | mIoU (%) | mean Class Accuracy (%) | mIoU (%) |
| Non-Linear | 100.00 | 100.00 | 99.97 | 99.91 |
| Affine | 100.00 | 100.00 | 100.00 | 100.00 |

Table 5: Analysis of Codebook Initialization to explain the performance advantage of the affine post-latent function.

| Decoder Function | Codebook Quantization | | |
| --- | --- | --- | --- |
| | K-means Inertia | Codebook Usage (%) | Quantization Accuracy (%) |
| Non-Linear | 17.838 | 100.00 | 98.71 |
| Affine | 0.944 | 100.00 | 100.00 |

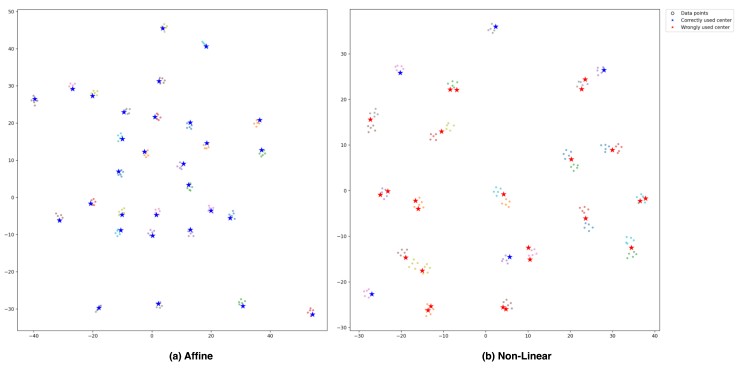

Figure 5: UMAP visualization of latent space quantization: the affine post-latent function enables correct alignment for codebook quantization, whereas the non-linear post-latent function leads to misaligned (incorrect) quantization.

**Projector in the Quantizer.** Our quantizer includes a learnable projector that adapts the fixed codebook to the encoder's current latent space during training. We ablate this component by removing the projector and directly computing distances in the original latent space. As shown in Tab.6, this results in codebook mismatch and reduced performance. The projector enables the model to

align quantization regions with the evolving latent geometry, especially when encoder parameters are updated during joint training.

Tab.7 summarizes the performance impact of all ablated settings, all evaluated under the information-theoretic lower bound.

Table 6: Ablation of the proposed Semantic Compression (single-token) on Geo-4Seg.

| Stage 1 | Stage 2 | mean Class Acc. (%) | mIoU (%) |
|---------|---------|---------------------|----------|
| ✓ | ✗ | 20.00 | 18.72 |
| ✗ | ✓ | 99.99 | 99.95 |
| ✓ | ✓ | 100.00 | 100.00 |

Table 7: Ablation of the proposed Semantic Compression (two-token) on Geo-4Seg.

| Stage 1 | Stage 2 | Stage 3 | mean Class Acc. (%) | mIoU (%) |
|---------|---------|---------|---------------------|----------|
| ✓ | ✗ | ✗ | 20.00 | 18.72 |
| ✗ | ✗ | ✓ | 20.00 | 18.72 |
| ✓ | ✓ | ✗ | 71.75 | 62.02 |
| ✗ | ✓ | ✓ | 83.82 | 83.22 |
| ✓ | ✓ | ✓ | 100.00 | 100.00 |

## 5 CONCLUSION

In this work, we introduce a new paradigm for image representation: *semantically optimal lossless compression*. Unlike prior representation learning frameworks, semantic compression explicitly targets losslessness and minimal code length with respect to a predefined set of downstream tasks.

We propose a concrete instantiation, **SOLO-VQ**, which employs a three-stage training procedure designed to jointly maximize codebook utilization and task accuracy. To enable rigorous evaluation, we construct a suite of synthetic datasets and task settings where optimality and losslessness can be analytically verified. We show that SOLO-VQ achieves both lossless task prediction and information-theoretic optimality on these synthetic benchmarks—something no existing method can accomplish, despite the simplicity of the domains.

We further demonstrate that SOLO-VQ generalizes to more complex scenarios, achieving lossless compression on previously unseen samples rather than merely memorizing training instances. In the multi-task setting with non-independent semantic targets, our method continues to achieve optimal lossless compression, matching the joint entropy lower bound.

A series of ablation studies validate the design choices of our method and highlight the importance of each component. Through these results, we establish the promise of SOLO-VQ as a new semantic representation paradigm. In future work, we aim to extend this framework to more complex, real-world datasets. We are optimistic that, given sufficiently diverse data and rich downstream tasks, SOLO-VQ can learn compact representations that generalize beyond the predefined task set.

## ETHICS STATEMENT

This work does not involve human subjects, personal or sensitive data, or potentially harmful applications. All contributions are intended solely for scientific research. We follow the ICLR Code of Ethics in all aspects of research design, experimentation, and presentation.

## REPRODUCIBILITY STATEMENT

All model details, training procedures, and hyperparameters are described in the paper and appendix. We will release code and data generation scripts upon acceptance.

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

# Appendix

## A  SYNTHESIZED DATASET

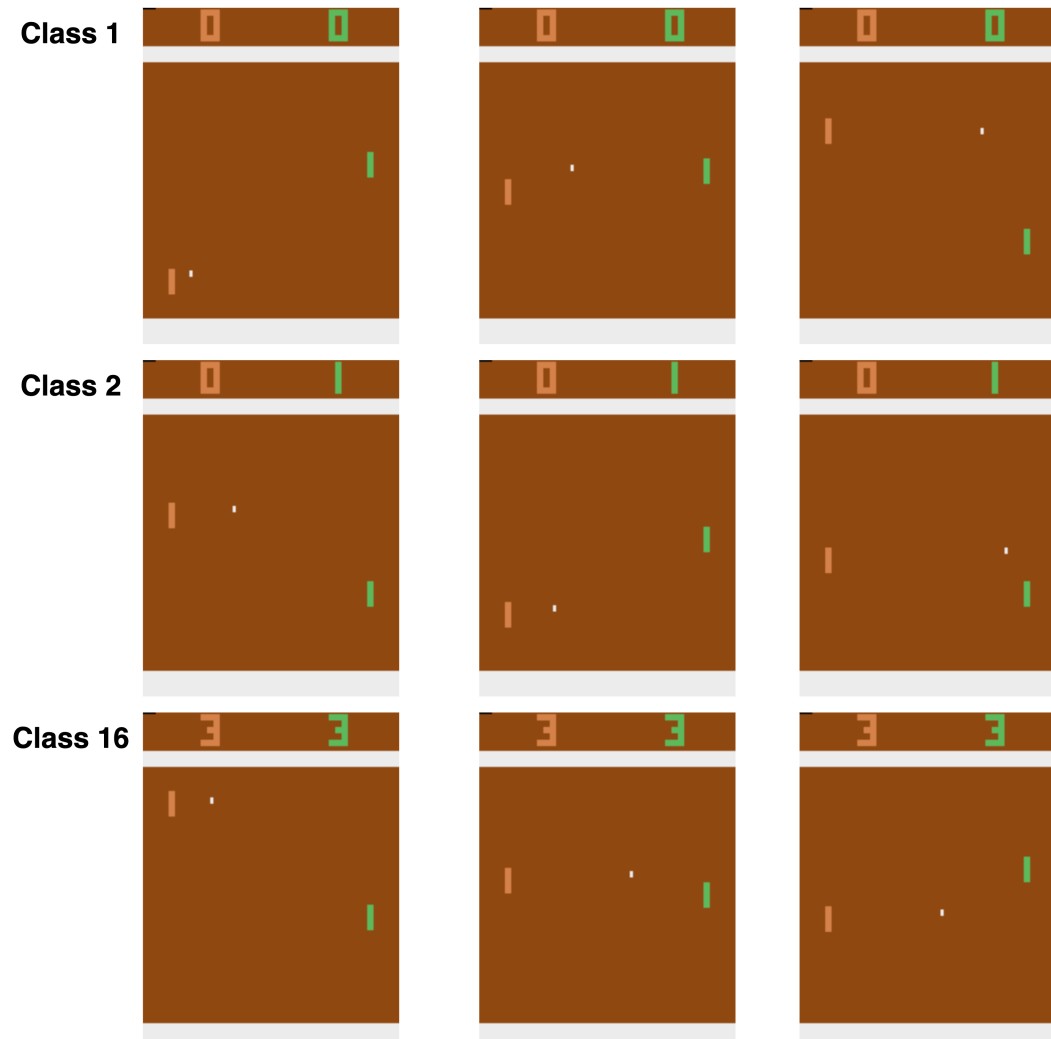

Figure 6: Pong-S: Score-pair classification. Lower-bound bitrate: 4 bits/image.

## B  CONVERGENCE-RATE COMPARISON

From the training curve 11 of our proposed SOLO-VQ and CVQ-VAE, we can notice that SOLO-VQ converges much faster than CVQ-VAE.

## C  PROJECTOR ARCHITECTURE

Table 8 compares MLP and Transformer architectures for the projector module. The experiments are performed in the two-token setting on PermRate-4Seg-A. To ensure fairness, both experiments start from the same encoder, codebook, and decoder weights, while only the projector architecture differs. Each projector is trained from scratch with a comparable number of parameters, enabling a controlled evaluation of the architectural impact.

The Transformer projector outperforms the MLP because self-attention enables each codeword to be refined in relation to all others. This global adjustment captures dependencies across codewords, resulting in a codebook structure that better aligns with encoder outputs and enhances quantization efficiency.

Table 8: Ablation Study: The model architecture for the Projector.

| Projector | # of Params. | mean Class Acc. (%) | mIoU (%) |
|---|---|---|---|
| MLP | 1.596 M | 98.11 | 97.98 |
| Transformer | 1.586 M | 100.00 | 100.00 |

## D  ALGORITHM PSEUDOCODE

---

**Algorithm 1** Initialize VQ Codebooks by Inertia-Threshold Selection

---

**Require:** Pretrained encoder $E_\theta$; training set $\mathcal{X} = \{x_i\}_{i=1}^n$; number of codebooks $C$; size options $S = \{1, 2, \ldots, N\}$; inertia threshold $\tau$ (default 25)

1: **Freeze** $\theta$
2: **Init representation pools:** $\mathcal{Z}^{(c)} \leftarrow \varnothing$ for all $c \in \{1, \ldots, C\}$
3: **for** each $x \in \mathcal{X}$ **do**
4:      $(z^{(1)}, \ldots, z^{(C)}) \leftarrow E_\theta(x)$           $\triangleright$ encoder outputs $C$ tokens
5:      **for** $c = 1$ **to** $C$ **do**
6:          append $z^{(c)}$ to $\mathcal{Z}^{(c)}$
7:      **end for**
8: **end for**
9: **for** $c = 1$ **to** $C$ **do**          $\triangleright$ one codebook per token position
10:     $\{(\mathbf{C}_s^{(c)}, I_s^{(c)})\}_{s \in S} \leftarrow \{\text{KMEANS}(\mathcal{Z}^{(c)}, k{=}s) \mid s \in S\}$   $\triangleright$ run K-means for all sizes, record inertia
11:     $s^{(c)} \leftarrow \min\{ s \in S \mid I_s^{(c)} \leq \tau \}$ **if exists; else** $s^{(c)} \leftarrow N$
12:     **Initialize codebook:** $\mathcal{C}^{(c)} \leftarrow \mathbf{C}_{s^{(c)}}^{(c)}$        $\triangleright$ centers $\Rightarrow$ codewords; size $= s^{(c)}$
13: **end for**
14: **return** $\{\mathcal{C}^{(c)}\}_{c=1}^C$          $\triangleright$ initialized codebooks only

---

**Algorithm 2** Progressive Codebook Shrinkage under Lossless Test Constraint

---

**Require:** **pretrained** encoder $E_\theta$ and decoder $D_\phi$; **randomly initialized** projectors $\{P_{\psi^{(c)}}\}_{c=1}^C$; codebooks $\{\mathcal{C}^{(c)}\}_{c=1}^C$ with sizes $\{s^{(c)}\}_{c=1}^C$ **initialized by Alg. 1**; size options $S{=}\{1, \ldots, N\}$; train set $\mathcal{X}_{\text{tr}}$, test set $\mathcal{X}_{\text{te}}$; test loss $\mathcal{L}_{\text{te}}$

1: **while** true **do**
2:     **Fine-tune model:** jointly update $(\theta, \phi, \{\psi^{(c)}\}, \{\mathcal{C}^{(c)}\})$ on $\mathcal{X}_{\text{tr}}$
3:     **Lossless test check:** compute $\mathcal{L}_{\text{te}}$ on $\mathcal{X}_{\text{te}}$; **if** $\mathcal{L}_{\text{te}} \neq 0$ **then return** $\{s^{(c)}\}_{c=1}^C$     $\triangleright$ stop—current sizes are the smallest achievable
4:     **Choose codebook to shrink (largest size):**
5:     **if** $\max_c s^{(c)} = 1$ **then return** $\{s^{(c)}\}_{c=1}^C$        $\triangleright$ all codebooks already at size 1
6:     **end if**
7:     $c^\star \leftarrow \arg\max_c s^{(c)}$        $\triangleright$ break ties by smallest $c$ (or any fixed rule)
8:     $s_{\text{next}} \leftarrow \max\{ s \in S \mid s < s^{(c^\star)} \}$        $\triangleright$ nearest smaller allowed size
9:     **Reinitialize codebook** $c^\star$ **at size** $s_{\text{next}}$**:**
10:    build pool $\mathcal{Z}^{(c^\star)} \leftarrow \{z^{(c^\star)} : (z^{(1)}, \ldots, z^{(C)}) \leftarrow E_\theta(x),\ x \in \mathcal{X}_{\text{tr}}\}$
11:    $(\mathbf{C}_{s_{\text{next}}}^{(c^\star)}, I_{s_{\text{next}}}^{(c^\star)}) \leftarrow \text{KMEANS}(\mathcal{Z}^{(c^\star)}, k{=}s_{\text{next}})$
12:    $\mathcal{C}^{(c^\star)} \leftarrow \mathbf{C}_{s_{\text{next}}}^{(c^\star)};\quad s^{(c^\star)} \leftarrow s_{\text{next}}$        $\triangleright$ centers become codewords
13:    **Continue loop.**
14: **end while**

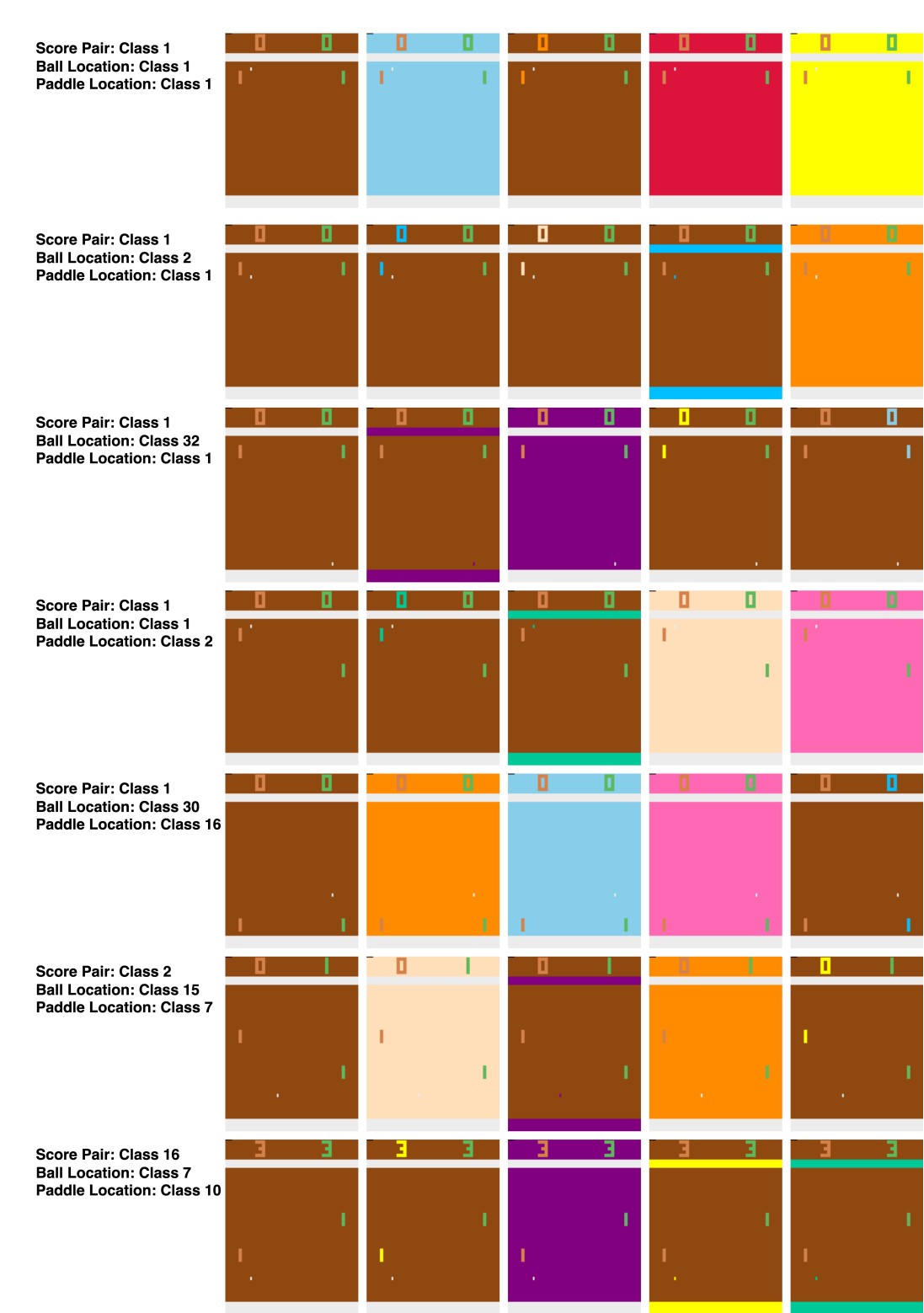

Figure 7: Pong-SPB: Multi-task classification of score pair, paddle location, and ball location. The three tasks are completely independent. Lower-bound bitrate: 13 bits/image.

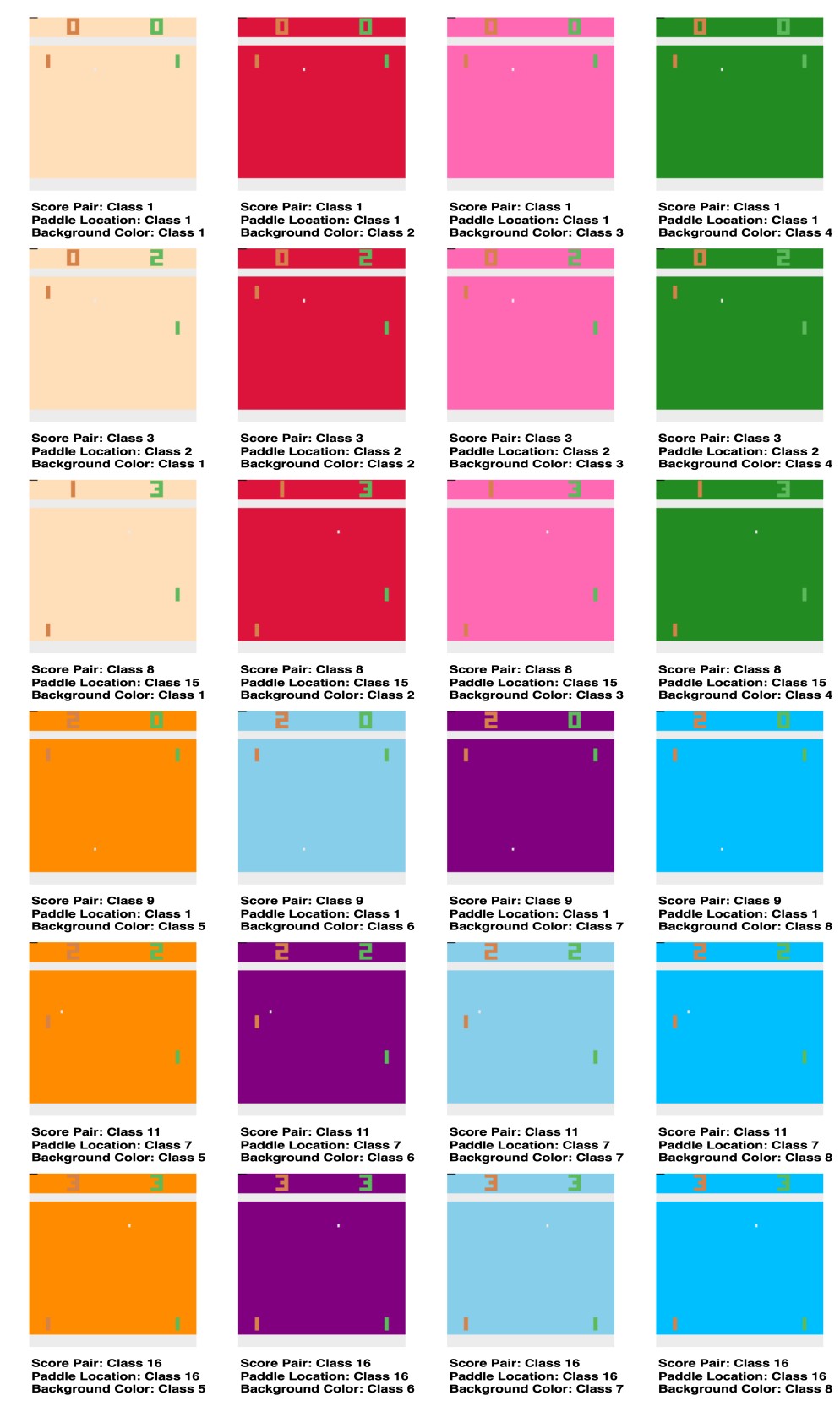

Figure 8: Pong-SPC: Multi-task classification of score pair, paddle location, and background color. The score-pair and background-color tasks are dependent: score pairs from classes 1–8 (0:0 to 1:3) are associated with only 4 of the 8 background colors, while classes 9–16 (0:0 to 1:3) use the remaining 4 colors, with no overlap. Both subsets are uniformly distributed across their respective colors. Lower-bound bitrate: 12 bits/image.

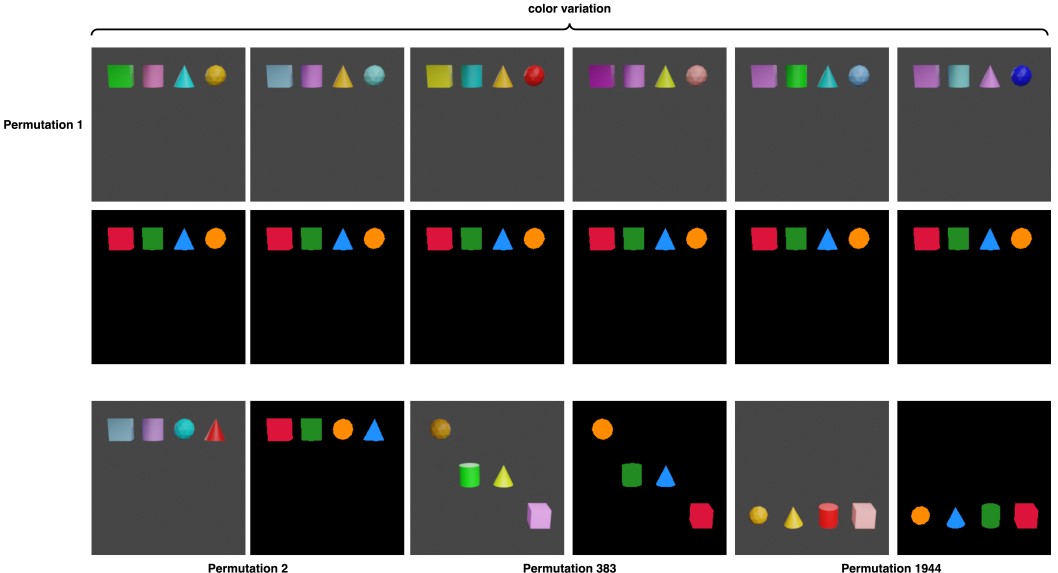

Figure 9: Geo-4Seg: Semantic segmentation of four objects and the background. Lower-bound bitrate: $\approx 10.925$ bits/image.

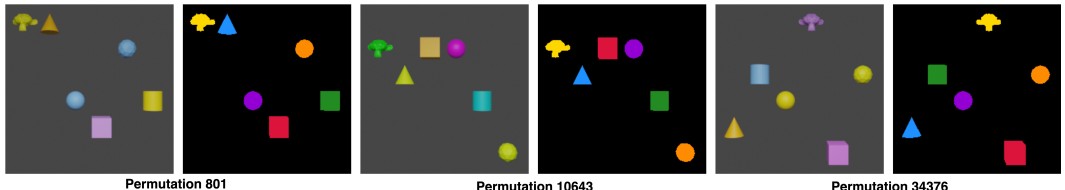

Figure 10: Geo-6Seg: Semantic segmentation of 6 objects and the background. Lower-bound bitrate: $\approx 25.002$ bits/image.

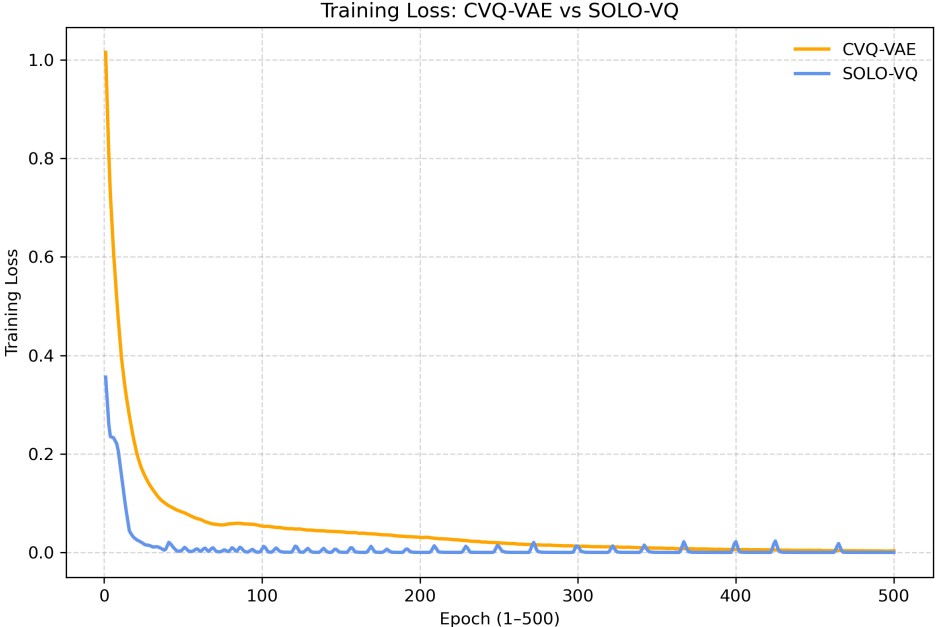

Figure 11: SOLO-VQ vs. CVQ-VAE on the training convergence rate.

