# OpenReview forum: "Semantic Optimal Lossless Vector Quantization"
_ICLR.cc/2026/Conference — ICLR 2026 Conference Withdrawn Submission_

### Official Review · Reviewer_DAus · 2025-10-28

**Soundness:** 1
**Presentation:** 2
**Contribution:** 2
**Rating:** 2
**Confidence:** 4

**Summary:**

The paper introduces a framework for learning discrete image representations that are both semantically lossless and information-theoretically optimal. The authors formalize semantic compression as encoding an image into a discrete code that preserves all information necessary for a set of downstream tasks, defining losslessness as perfect task recoverability and optimality as matching the joint entropy of the task variables. They propose a three-stage training pipeline: 1) latent pretraining 2) clustering-based codebook initialization 3) projector-based codebook adaptation. They evaluate it on synthetic datasets (Pong and geometric segmentation tasks) where entropy bounds are analytically computable. SOLO-VQ achieves perfect reconstruction of task semantics and minimal bitrates close to theoretical lower bounds, outperforming baselines such as VQ-VAE.

**Strengths:**

The paper contributes a formalization of “semantic lossless optimal compression,” a conceptually clean unification of information theory and task-based representation learning.

The idea of testing optimality in synthetic, fully enumerable domains enables the authors to make theoretically grounded claims that are rarely possible in learned compression.

The motivation and task description are clear.

**Weaknesses:**

The paper does not discuss or cite relevant prior work on compression optimized for downstream tasks, such as recognition-aware or task-aware compression methods [1, 2]. This omission weakens the positioning of SOLO-VQ within existing literature, as comparisons are only made against other vector-quantization techniques that pursue a different objective.

Experiments are restricted to small, synthetic datasets with enumerable task spaces, which severely limits the conclusions. There is no evidence that the proposed method scales to natural images or continuous semantics, and the claim of “semantic optimality” remains untested in realistic settings.

The approach assumes complete knowledge of all downstream tasks, preventing generalization beyond the predefined set.

Presentation quality is poor. Several tables simply report 100 % accuracy across all configurations, providing no meaningful insight. These could be summarized in a single sentence rather than repeated in multiple tables.

Key baselines (VQ-STE++, CVQ-VAE, FSQ) lack citations or clear experimental details, making reproducibility difficult and casting doubt on the fairness of the comparisons.

Section 2.1, the second item in the list has almost the same sentence twice.

[1] Kawawa-Beaudan, Maxime, Ryan Roggenkemper, and Avideh Zakhor. "Recognition-aware learned image compression." arXiv preprint arXiv:2202.00198 (2022).
[2] Choi, Jinyoung, and Bohyung Han. "Task-aware quantization network for jpeg image compression." European Conference on Computer Vision. Cham: Springer International Publishing, 2020.

**Questions:**

Can SOLO-VQ scale to real datasets where entropy is not analytically computable? Results on MNIST or CIFAR-10 would help validate whether semantic optimality approximately holds in practice.

Include comparisons with task-aware compression methods, which also optimize for classification or detection accuracy.

Finally, add missing citations and details for VQ-STE++, CVQ-VAE, and FSQ to ensure reproducibility and fair comparison.

---

### Official Review · Reviewer_fDcq · 2025-10-29

**Soundness:** 3
**Presentation:** 2
**Contribution:** 3
**Rating:** 4
**Confidence:** 3

**Summary:**

The paper tries to formalize semantic lossless optimal compression. It proposes that a discrete code should preserve all information needed to solve a predefined set of tasks with zero performance loss, and its entropy should match the joint entropy of the task labels as the theoretical lower bound.
It proposes SOLO-VQ, a three-stage pipeline including VQ-aware latent pretraining, k-means codebook initialization, and projector-based codebook adaptation. It aimed at maximizing codebook utilization and task accuracy under a rate constraint.

**Strengths:**

1. The method provably hits entropy lower bounds on synthetic tasks.
2. Using synthetic datasets where task entropies are computable, the authors show perfect task performance at the lower-bound bitrate.
3. The method shows some generalization and rate-distortion efficiency on Geo-6Seg.
4. It supports multi-task composition with independent/conditional targets.

**Weaknesses:**

1. The evaluation in this paper is restricted to synthetic datasets with computable entropy.
2. The paper should discuss about the scalability of codebooks and corresponding training complexity.
3. There is no discussion and comparisons about ImageNet-style vision tasks besides only the MSE distortion.
4. The semantics in the examined datasets is quite simple. The performance of the proposed methods and affine heads' capacity on complex semantics should be discussed.

**Questions:**

1. How would you approximate joint task entropy on real-world datasets to claim "near-optimality" without closed-form counts?
2. Is that possible for you to bound performance drop for unseen tasks?
3. Is there any experimental results on ImageNet-style tasks where the distortion is task error rather than pixel MSE?

---

### Official Review · Reviewer_HYvh · 2025-10-31

**Soundness:** 2
**Presentation:** 3
**Contribution:** 2
**Rating:** 4
**Confidence:** 3

**Summary:**

This paper introduces Semantic Optimal Lossless Vector Quantization (SOLO-VQ), a framework designed to create a compressed representation of an image that perfectly preserves all information for a predefined set of downstream tasks while minimizing code length. The method's claims of achieving "optimal" and "lossless" compression are validated on synthetic datasets where the information-theoretic bounds are analytically computable.

**Strengths:**

The paper does a good job of formally defining "semantic lossless" and "semantic optimal" compression. This provides a clear, principled language for discussing task-oriented representation learning, even if the underlying concept is not entirely new.

**Weaknesses:**

1.The paper's conclusions are exclusively drawn from "toy" synthetic datasets (Pong, Geo). These environments are fully discrete, noise-free, and lack the complexity of real-world visual data. Demonstrating "optimality" in such a sterile setting provides little to no evidence that the approach would be viable or even meaningful on realistic datasets like ImageNet. The claims of success are therefore not generalizable and potentially trivial given the simplicity of the tasks.
2.The proposed three-stage SOLO-VQ method is a straightforward composition of existing techniques in the vector quantization literature (e.g., k-means initialization, use of projectors to bridge latent spaces). The method does not introduce a novel algorithmic or architectural contribution but rather reassembles known components for its specific goal.

**Questions:**

Could you clarify how your "semantic compression" framework fundamentally differs from a discrete version of the Information Bottleneck principle, where the goal is to learn a compressed variable Z that maximizes mutual information with the target labels I(Z;Y) while minimizing information from the input I(Z;X)?

---

### Official Review · Reviewer_asYb · 2025-11-01

**Soundness:** 2
**Presentation:** 3
**Contribution:** 3
**Rating:** 4
**Confidence:** 3

**Summary:**

This paper propose a new semantic image compression method based on semantic optimal lossless vector quantization, to preserve semantic information without performance loss for a class of downstream tasks.

**Strengths:**

* It proposes a novel paradigm of semantic image compression and defines several new concepts.
* The use of synthetic datasets with computable entropy limits is a good contribution, enabling empirical proof of optimality, which is rarely verifiable in prior work.
* It is reported that SOLO-VQ achieve both lossless task prediction and information-theoretic optimality on proposed synthetic benchmarks.

**Weaknesses:**

* The evaluation is confined to simple synthetic datasets, leaving the performance of proposed method on complex, continuous real-world images entirely unexplored and uncertain.
* Comparisons are made primarily against older Vector Quantization models, lacking benchmarks against modern state-of-the-art representation learning methods.
* The relaxed definition of "lossless" as perfect accuracy on a test set is achievable only in synthetic environments and does not translate to real-world scenarios where 100% accuracy is impossible.
* Fig. 1 on page 2 is not corresponding to the texts, as the T_n is missing in the picture.
* The presentation needs further improving. Although the background of image compression is familiar to those who are major in coding, more information about related works is necessary.

**Questions:**

Please refer to the weaknesses.

---

### Note · Authors · 2025-11-20

**Comment:**

We sincerely appreciate everyone’s time and the detailed reviews and recommendations. We will thoroughly review them and incorporate the necessary improvements.

**Withdrawal Confirmation:**

I have read and agree with the venue's withdrawal policy on behalf of myself and my co-authors.